# Liquid-liquid phase separation mediated immune evasion of respiratory syncytial virus against oligoadenylate synthetase-RNase L pathway

Woo Yeon Hwang[1,2], Michael G. Rosenfeld[3], Soohwan Oh[2]*, Young-Chan Kwon[1,4]*

1 Center for Infectious Diseases Vaccine and Diagnosis Innovation (CEVI), Korea Research Institute of Chemical Technology (KRICT), Daejeon, Republic of Korea, 2 College of Pharmacy, Korea University, Sejong, Republic of Korea, 3 Department and School of Medicine, University of California, San Diego, California, United States of America, 4 Medical Chemistry and Pharmacology, University of Science and Technology (UST), Daejeon, Republic of Korea

* soohwanoh@korea.ac.kr (SH); yckwon@krict.re.kr (YCK)

## Abstract

Respiratory syncytial virus (RSV) infection is the major cause of severe respiratory illnesses in infants and older adults. RSV forms phase-separated biomolecular condensates called inclusion bodies (IBs), which serve as hubs for viral replication. However, the contribution of IBs to host immune response evasion remains elusive. We report that RSV IBs protect viral RNA from the 2′-5′ oligoadenylate synthetase (OAS)-RNase L pathway, a critical antiviral defense mechanism that cleaves viral and cellular RNAs. RSV infection did not activate the OAS-RNase L pathway, and ectopically activated RNase L did not suppress viral replication. In RSV-infected cells, double-stranded RNA (dsRNA) was efficiently sequestered within liquid–liquid phase separation (LLPS)-mediated IBs, rendering its detection challenging. LLPS perturbation caused dsRNA release from IBs into the cytosol. dsRNA extracted from infected cells, which lacked LLPS shielding, triggered OAS-RNase L pathway activation. Thus, LLPS-driven IBs structurally sequester viral RNA, facilitating RSV to evade RNase-dependent genomic RNA degradation mediated by the OAS-RNase L antiviral pathway.

## Author summary

Biomolecular condensates are increasingly recognized as key mechanisms of subcellular organization. The role of viral infection-mediated phase-separated structures, called inclusion bodies (IBs), in replication has already been characterized in several viruses, including the human respiratory syncytial virus (RSV). Herein, we focused on the role of IBs in immune evasion and found that RSV uses inclusion bodies to compartmentalize viral RNA and escape detection via the OAS–RNase L antiviral defense pathway.

---

**Data availability statement:** All relevant data are within the manuscript and its Supporting Information files.

**Funding:** This work was supported by the Korea Research Institute of Chemical Technology (KRICT) under Project No. KK2633-20 (A Study on the Next-Generation Infectious Disease Control Technology Program to Y.-C.K.). Y.-C. K was also supported by a National Research Foundation of Korea (NRF) grant funded by the Ministry of Education, Science and Technology (RS-2023-00208568). SO was supported by the Basic Science Research Program through the National Research Foundation of Korea (NRF), funded by the Ministry of Education (2022R1C1C1010699) and by the Ministry of Science and ICT (MSIT), Korea, under the Information Technology Research Center (ITRC) support program (IITP-2025-RS-2023-00258971), supervised by the Institute for Information & Communications Technology Planning & Evaluation (IITP). The funders had no role in study design, data collection and analysis, decision to publish, or preparation of the manuscript.

**Competing interests:** The authors have declared that no competing interest exists.

## Introduction

Membrane-less organelles (MLOs) such as inclusion bodies (IBs) and nuclear bodies are dynamic cellular compartments present in different cell regions. These structures are formed via liquid–liquid phase separation (LLPS) of biomolecules, RNAs, and proteins with intrinsically disordered regions [1–3]. Unlike traditional organelles, MLOs lack physical barriers, facilitating the reversible and dynamic assembly of distinct liquid-like droplets that can be visualized under a microscope [4–6]. This unique property is exploited by viruses to improve replication efficiency, evade host immunity, and remodel cellular environments, making them the main therapeutic targets for antiviral strategies [7].

Respiratory syncytial virus (RSV) is a major cause of acute respiratory tract infections worldwide, with most people being exposed to the virus by the age of two [8,9]. While RSV typically causes mild symptoms resembling those of the common cold, it can lead to severe diseases such as bronchiolitis and pneumonia in vulnerable populations, including infants, older adults, and immunocompromised individuals [10,11]. Effective vaccines for direct administration to infants remain unavailable, despite recent advancements in RSV vaccine development, including regulatory approvals for vaccines aimed at older adults and maternal immunization strategies [12–18]. RSV is an enveloped, negative-sense, single-stranded RNA virus belonging to the *Orthopneumovirus* genus of the *Pneumoviridae* family [19]. It has a non-segmented RNA genome of approximately 15 kb in length and encodes 10 genes that are translated into 11 proteins. Among these viral proteins, nucleoprotein (N), phosphoprotein (P), polymerase (L), and the transcription factor M2-1 accumulate in the IBs (MLO subtype) that form in the cytoplasm of RSV-infected cells [20–24]. These IBs are generated through LLPS and serve as hubs for viral RNA synthesis, with newly synthesized viral RNA concentrated within substructure IB-associated granules (IBAGs) embedded in IBs via M2-1–mediated trafficking. [5,25–27]. Notably, LLPS hardening blocks RSV replication with disorganization of IBAGs, underscoring the functional criticality of phase separation in the viral life cycle [28]. While IBs are well-recognized as the site supporting viral RNA synthesis, emerging evidence suggests that they may also contribute to immune evasion, possibly by leveraging their dynamic properties to compartmentalize viral RNA and shield it from host defenses [29,30]. However, these additional functions are yet to be completely elucidated.

Double-stranded RNA (dsRNA) is an inevitable byproduct of viral replication, particularly in negative-sense single-stranded RNA viruses such as RSV, in which copy-back defective viral genomes (cbDVGs), generated during viral genome replication, represent a major dsRNA species [31]. Almost all organisms exhibit pathogen recognition receptors capable of detecting dsRNA and initiating antiviral immune responses [32]. The 2′-5′ oligoadenylate synthetase (OAS) system induced by the interferon (IFN) signaling pathway is one such sensor that detects dsRNA. OASs become catalytically active upon binding to dsRNA and synthesize 2′-5′ oligoadenylates (2-5A) from ATP molecules, acting as secondary messengers to activate ribonuclease L (RNase L). Activated RNase L dimerizes and cleaves cellular and viral single-stranded RNA, resulting in the indiscriminate suppression of protein

expression from viral and host sources [33,34]. The OAS-RNase L pathway has been confirmed to exert antiviral activity against different viruses, including the Sindbis virus, West Nile virus, dengue virus, and SARS-CoV-2 [35–38], whereas some viruses, such as influenza and Zika viruses (ZIKV), use diverse strategies to evade this antiviral mechanism [39–42]. However, the antiviral effects of the OAS-RNase L pathway on RSV have not yet been clearly established.

Herein, we investigated the mechanism by which RSV evades the antiviral activity of the OAS-RNase L pathway. By measuring viral RNA and ribosomal RNA (rRNA) cleavage, we found that RSV efficiently circumvents the host dsRNA-sensing mechanism, including OAS activation, by forming IBs driven by LLPS. Using dsRNA-specific antibodies, we observed that viral RNA was sequestered within the IBs, protecting them from recognition and degradation by activated RNase L. These findings show that LLPS-driven IBs serve as replication hubs and protective compartments, facilitating RSV to escape RNase-dependent degradation of genomic RNA mediated by the OAS-RNase L antiviral pathway.

## Results

### RSV infection does not activate the OAS-RNase L pathway

The OAS-RNase L pathway is activated against infections from a diverse range of viruses [35–38]. We infected the A549 cells with RSV and assessed RNase L activation via rRNA cleavage, the hallmark of RNase L activation, at various times after infection to investigate whether RSV infection activates the OAS-RNase L pathway. We used the prototypic strains RSV A2 and RSV 18537, which represent historically established reference strains for the RSV subgroups A and B. We transfected cells with polyinosinic:polycytidylic acid (poly (I:C)) as a surrogate for dsRNA to activate the OAS-RNase L pathway as a positive control [35]. Despite efficient viral replication over time, rRNA cleavage was not observed in the cells infected with either RSV strain, regardless of the time point (Fig 1A and 1B). Also, rRNA cleavage was not observed in the HEp-2 cells, which are highly susceptible to RSV infection (S1A and S1B Fig). By contrast, rRNA cleavage was readily detected over time in ZIKV-infected cells, consistent with previous reports (S1C and S1D Fig) [39]. Furthermore, RSV infection with different multiplicities of infection (MOI), ranging from 0.1–5, did not induce rRNA cleavage (Fig 1C). Next, we determined the protein and mRNA expression levels of genes related to the OAS-RNase L pathway in the RSV-infected cells (Fig 1D and 1E). As expected, *OAS1*, *OAS2*, and *OAS3*, which are the components of interferon-stimulated genes, were upregulated after RSV infection, consistent with increased *IFNβ* expression.

Activated OAS3 and RNase L are known to form biomolecular condensates upon dsRNA sensing [43,44]. To further investigate the subcellular dynamics of this pathway during RSV infection, we examined OAS3 condensate formation using A549 RNase L-KO cells stably expressing GFP-RNase L and mRuby2-OAS3. In mock-infected cells, OAS3 was diffusely distributed throughout the cytoplasm; however, upon poly (I:C) transfection, OAS3 rapidly concentrated into discrete condensates that became increasingly apparent over time (Fig 1F). In striking contrast, RSV-infected cells showed OAS3 localization patterns indistinguishable from mock-infected cells, with no temporal changes in localization or condensate formation.

The distinct OAS isoforms generated by alternative splicing exhibit different antiviral activity against diverse viruses [35,37]. Moreover the A549 cells do not produce the OAS1 p46 isoforms, as shown previously [38]. To enhance A549 cell responsiveness to dsRNA, various isoforms, including OAS1 p46, were overexpressed before RSV infection; however, RSV infection still did not induce rRNA cleavage (S2 Fig). Thus, RSV infection did not activate the OAS-RNase L pathway despite the upregulation of the expression levels of its components.

### RSV circumvents OAS-RNase L pathway activation

Ectopically activated RNase L suppresses viral replication, even in the absence of OAS-RNase L pathway activation via viral infection [39,45]. Thus, we aimed to examine the effect of ectopically activated RNase L on RSV replication since RSV infection does not activate the OAS-RNase L pathway. Cells were infected with RSV for 24 h and subsequently

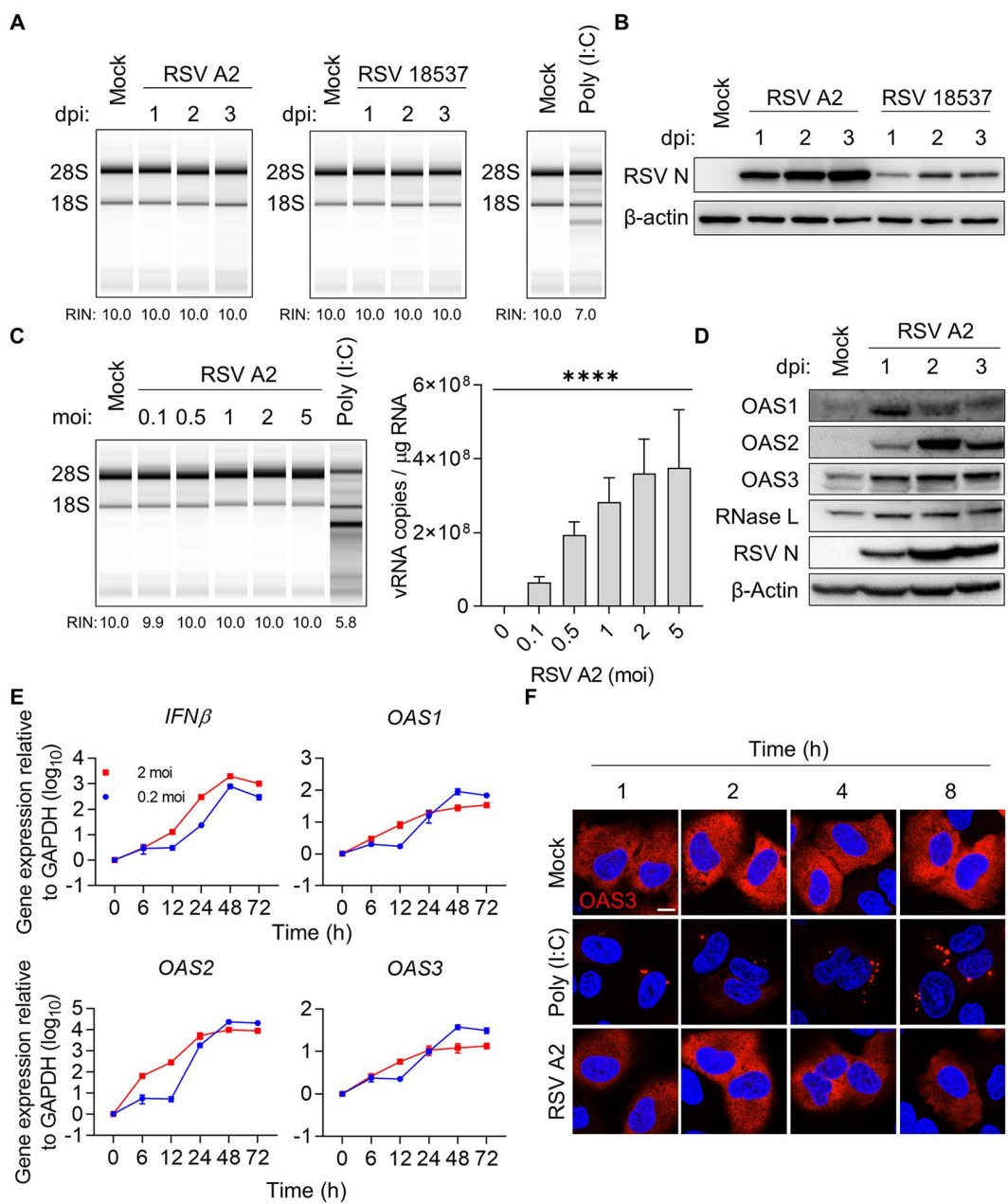

**Fig 1. RSV infection does not elicit OAS-RNase L activation. (A)** A549 cells were infected with RSV A2 or RSV 18537 at an MOI of 2. Cells were collected at indicated time points. rRNA cleavage and RNA integrity number (RIN) were analyzed using an RNA TapeStation System. **(B)** Protein levels were examined using immunoblotting with anti-RSV N and anti-β-actin antibodies. **(C)** A549 cells were infected with RSV A2 at the indicated multiplicities of infection (MOIs) and lysed 2 d after-infection. Viral RNA levels were quantified using quantitative reverse transcriptase-polymerase chain reaction (RT-qPCR). **(D)** After RSV infection at an MOI of 2, protein levels of the OAS-RNase L pathway components (OAS1, OAS2, OAS3, and RNase L) were analyzed at the indicated time points via immunoblotting. (E) mRNA levels related to the IFN-β and OAS-RNase L pathway in the RSV-infected cells were measured using RT-qPCR and normalized to *GAPDH*. **(F)** A549-RNase L KO stably expressing GFP-RNase L and mRuby2-OAS3 cells were infected with RSV A2 or transfected with poly (I:C) and imaged using an LSM 980 confocal microscope. Scale bar, 10 µm. Data represent mean ± standard errors of the mean from three independent experiments. Statistical significance was determined via one-way analysis of variance **(C)**; ****P < 0.0001.

transfected with poly (I:C) to activate RNase L. rRNA cleavage was observed in all poly (I:C)-transfected cells, regardless of RSV infection (Fig 2A). These observations showed that RSV does not impede the OAS-RNase L pathway activated by ectopic dsRNA. Notably, RNase L, activated by different poly (I:C) doses, did not inhibit RSV replication (Fig 2B). Moreover, although OAS3 condensates were not detected in RSV-infected cells (Fig 1F), OAS3 nonetheless formed condensates when ectopically activated by poly (I:C) in RSV-infected cells (Fig 2C); however, these condensates did not co-localized with the RSV N protein.

Overall, these results indicate that RSV viral RNA generated during replication evades OAS-mediated dsRNA-sensing without impairing that pathway, and is likely protected from degradation by the activated OAS-RNase L pathway.

## Structural masking of viral dsRNA within IBs underlies RSV's evasion of cytosolic sensing

During viral replication, dsRNA is generated, although RSV infection does not activate dsRNA-dependent OASs. The dsRNA and viral protein (F, RSV fusion protein) were probed using specific antibodies in an immunofluorescence assay (IFA) in RSV-infected cells for 48 h to evaluate the extent of dsRNA production during RSV replication (Fig 3A). Surprisingly, dsRNA was not detected, although the viruses efficiently replicated, as indicated by viral protein (F) expression. Contrastingly, ZIKV-infected cells showed clear dsRNA signals along the viral protein (E, ZIKV envelope protein). Because dsRNA is inevitably produced during the RNA virus replication process and is a known target for the host defense system, considering its absence in virus-infected cells is illogical [32]. We resolved this contradiction by conducting an experiment involving immunoblotting to probe for dsRNA in total RNA samples isolated from RSV-infected cells using a previously reported methodology (S3A Fig) [46]. The dsRNA signal was detected in the RSV-infected samples (Fig 3B), and was further confirmed using another anti-dsRNA antibody (9D5) targeting a different epitope (S3B Fig).

RSV infection induces the formation of cytoplasmic IBs that serve as replication compartments [20,21,25]. Immunofluorescence staining confirmed IBs formation as early as 12 h post-infection, with progressive expansion over time (S4A Fig). IBs were enriched with viral replication proteins (N, P, M2-1, and L protein) and newly synthesized viral RNA, confirming their role as active replication sites (S4B Fig) [22–24]. The minimal components necessary for IBs formation, N and P proteins, displayed an uneven distribution within the hollow-appearing IBs, a pattern that became more pronounced by 24

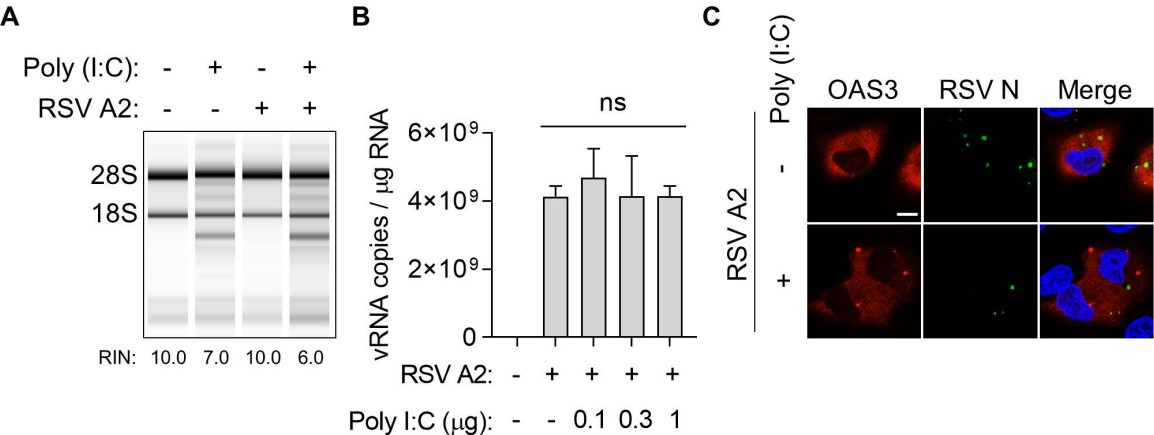

**Fig 2. RSV evades OAS–RNase L pathway rather than inhibits. (A, B)** A549 cells infected with RSV A2 at an MOI of 2 were transfected with (A) 1 μg or (B) indicated amount of poly (I:C) at 24 h post-infection. RNA was isolated 24 h later and analyzed for rRNA cleavage and viral RNA levels. **(C)** A549-RNase L KO stably expressing GFP-RNase L and mRuby2-OAS3 cells were transfected with poly (I:C) or mock at 24 h post-infection with RSV A2, and imaged by confocal microscopy. Scale bar, 10 μm. Statistical significance was determined using a one-way analysis of variance **(B)**; ns: not significant. Data represent mean ± standard errors of the mean.

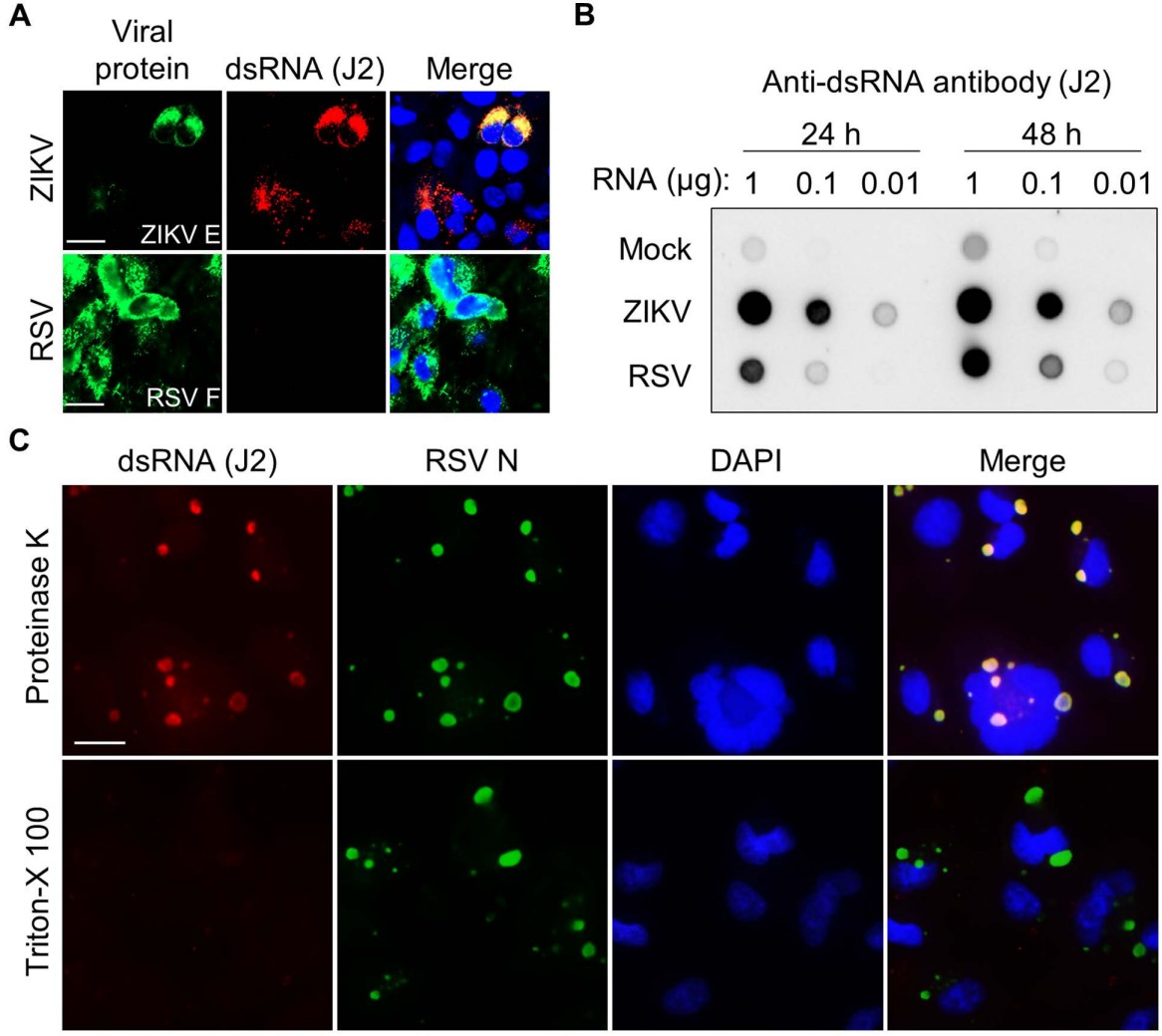

**Fig 3. Sequestering dsRNA within inclusion bodies (IBs). (A)** A549 cells infected with RSV A2 (MOI = 2) were fixed at 2 d post-infection and stained with anti-dsRNA (J2, red) and anti-RSV F antibodies (green). ZIKV-infected cells (MOI = 2) stained with anti-ZIKV envelope antibody (green). Scale bar: 20 μm. **(B)** Total RNA from mock-, RSV-, or ZIKV-infected cells was transferred to a nylon membrane. dsRNA was detected using anti-dsRNA antibody (J2). Images were acquired using an LSM 980 confocal microscope. Scale bar, 20 μm. **(C)** RSV-infected cells at 24 h post-infection were treated with proteinase K before staining. Images were obtained using a Nuance FX Multispectrum Imaging System. Scale bar, 20 μm.

h post-infection (S4A and S4B Fig), as repeatedly reported [25,26]. These structural characteristics led us to hypothesize that IBs restrict antibody accessibility to dsRNA. To test this, we treated cells with proteinase K during permeabilization—a standard procedure in fluorescence in situ hybridization to remove RNA-binding proteins [47]. Proteinase K treatment enabled detection of dsRNA within IBs, where it co-localized with N protein but was excluded from IBAGs, consistent with the previously reported localization of RSV genomic RNA (Fig 3C) [25]. This finding demonstrates that dsRNA is structurally masked by the IBs, explaining the discrepancy between IFA and immunoblotting results (Fig 3A and 3B). In RSV-infected cells, both OAS3 and RNase L were excluded from IBs (S5A and S5B Fig). Given that ectopically activated OAS3 condensates did not co-localize with IB-sequestered dsRNA (Fig 2C), these observations indicate that IBs exclude OAS–RNase L components from IBs where dsRNA accumulates during RSV replication. Collectively, these findings suggest

that IBs function as hubs for viral RNA synthesis while simultaneously sequestering dsRNA by-products in a structurally concealed manner that limits OAS–RNase L–mediated sensing and degradation.

## Disruption of LLPS in RSV IBs releases dsRNA into the cytosol

The morphology and properties of IBs formed during RSV replication suggest they are condensates generated through LLPS [26–28]. To disrupt these structures, RSV-infected cells were treated with 5% 1,6-hexanediol (1,6-HD), a phase separation disruptor, or subjected to hypotonic shock [5,28,48]. Following 1,6-HD or hypotonic shock treatment, dsRNA was progressively released from IBs into the cytosol (Fig 4A and 4B). We confirmed that neither treatment induced host dsRNA production, and 1,6-HD-induced dsRNA release was validated using an alternative anti-dsRNA antibody (9D5) co-stained with P protein (S6 and S7A Figs). A portion of the dsRNA was observed at the periphery of the IBs in the early stages following 1,6-HD exposure (Fig 4A, white arrow and white box), a pattern similarly observed under hypotonic conditions with moderate intensity (Fig 4B, white arrow and white box).

To validate these observations in a physiologically relevant context, we examined IBs formation and dsRNA release in primary normal human bronchial epithelial (NHBE) cells, observing comparable patterns following 1,6-HD or hypotonic shock treatment (Fig 5 and S7B Fig). These findings demonstrate that RSV IBs sequester dsRNA via LLPS, preventing OAS-RNase L pathway surveillance during viral replication.

## Naked dsRNA isolated from RSV-infected cells induces OAS-RNase L pathway activation

To assess activation of the OAS–RNase L pathway following LLPS disruption and dsRNA release, RSV-infected cells were treated with 5% 1,6-HD or subjected to hypotonic shock, followed by recovery for 24 h. In both cell lines (A549 and

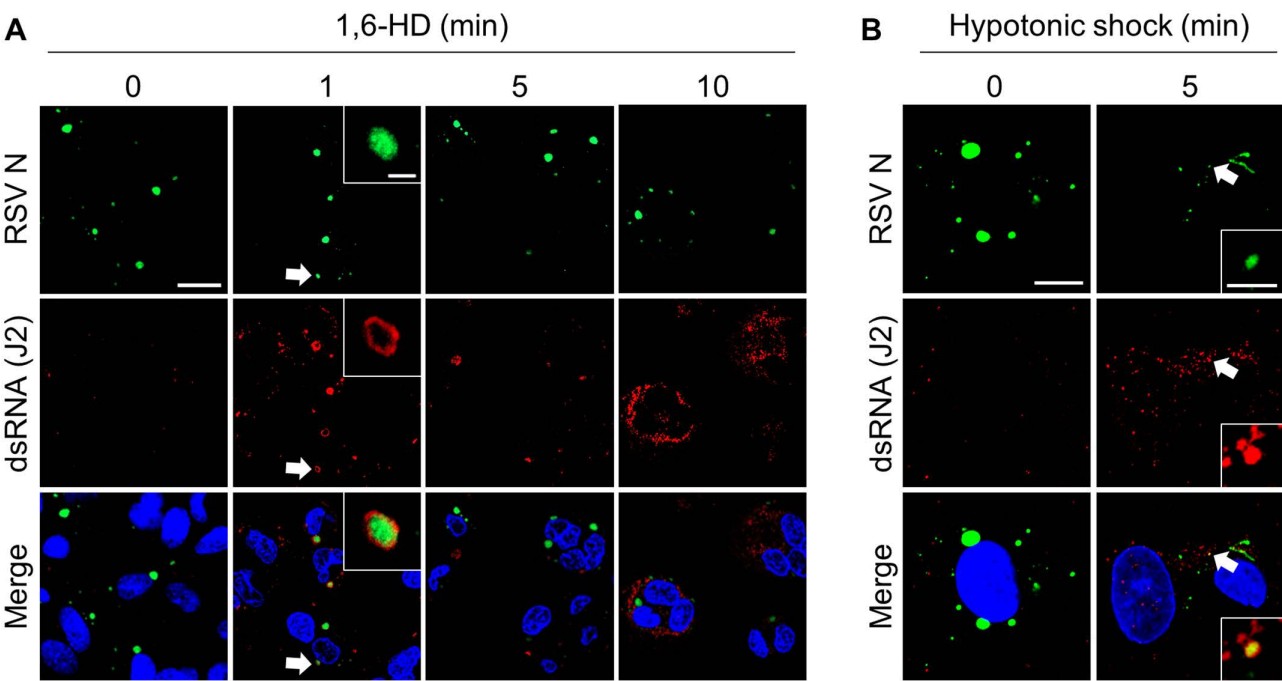

**Fig 4. Inclusion bodies (IBs) disruption induces dsRNA leakage. (A, B)** A549 cells were infected with RSV A2 (MOI = 2) for 24 h and then exposed to 5% 1,6-hexanediol (1,6-HD) or hypotonic shock for the indicated durations prior to fixation. Cells were stained with anti-N antibody (green) and anti-dsRNA (J2, red). Scale bar, 10 μm. The boxed area are shown magnified IBs indicated by white arrow. Scale bar, 2 μm.

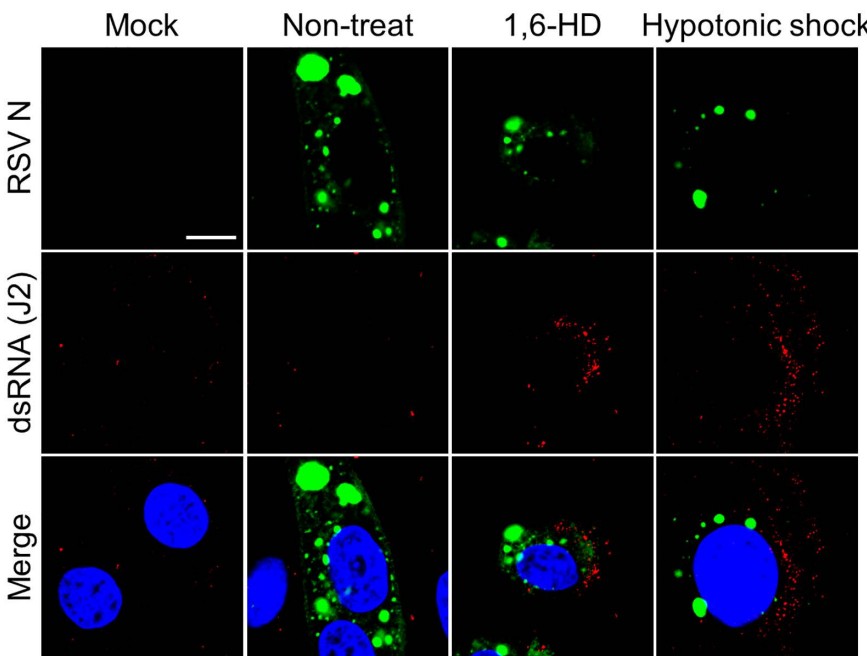

**Fig 5. RSV dsRNA leakage induced by LLPS disruption in normal human bronchial epithelial (NHBE) cells.** NHBE cells infected with RSV A2 at an MOI of 2 were exposure to 5% 1,6-hexanediol (1,6-HD) or hypotonic shock at 24 h post-infection, and then stained with anti-RSV N antibody (green) and anti-dsRNA (J2, red). Scale bar, 10 μm.

NHBE), dsRNA release induced by 1,6-HD or hypotonic shock did not result in detectable rRNA cleavage (S8A and S8B Fig).

To exclude the barrier function of LLPS toward RSV-derived dsRNA and to investigate the dsRNA-sequestering role of IBs, naked dsRNA was isolated from RSV-infected cells using a pull-down assay with a dsRNA-specific antibody, as previously described (Fig 6A) [49]. Isolated dsRNA was confirmed via dot blot assay using anti-dsRNA antibodies (Fig 6B and S9B Fig). Subsequently, we observed significant rRNA cleavage after the transfection of purified naked dsRNA from RSV-infected cells, whereas dsRNA from mock-infected cells did not induce cleavage. (Fig 6C). This trend was consistently observed when total RNA from RSV- or mock-infected cells was used for transfection (S9A Fig). Thus, dsRNA, generated during RSV replication, is protected by IBs, even though it is detectable via the OAS-RNase L pathway.

## Discussion

The OAS-RNase L pathway, an IFN-stimulated antiviral mechanism, is crucial for the innate immune defense against viral infections [33,34]. This pathway is initiated following the detection of dsRNA produced during RNA virus replication via OASs. Additionally, activated RNase L indiscriminately degrades viral and host RNA, suppressing viral replication [35–38]. Contrastingly, viruses use diverse strategies to evade this pathway [39,41,42,50]. The relationship between RSV infection and the OAS-RNase L pathway has been reported in several studies [51–54]. However, the precise molecular mechanisms involved remain to be completely elucidated. Herein, we showed that RSV evades the OAS-RNase L pathway by sequestering viral RNA within LLPS-driven IBs, preventing OASs from detecting dsRNA. Consequently, even ectopically activated RNase L fails to suppress RSV replication. RSV utilizes multiple complementary strategies to target interconnected immune responses: NS1 and NS2 proteins antagonize type I IFN production and signaling; the N protein sequesters host immune factors such as MDA5 and MAVS within IBs [29,55]. This immune evasion mechanism has significant physiological implications for RSV pathogenesis in the respiratory tract, where airway epithelial cells serve as the primary

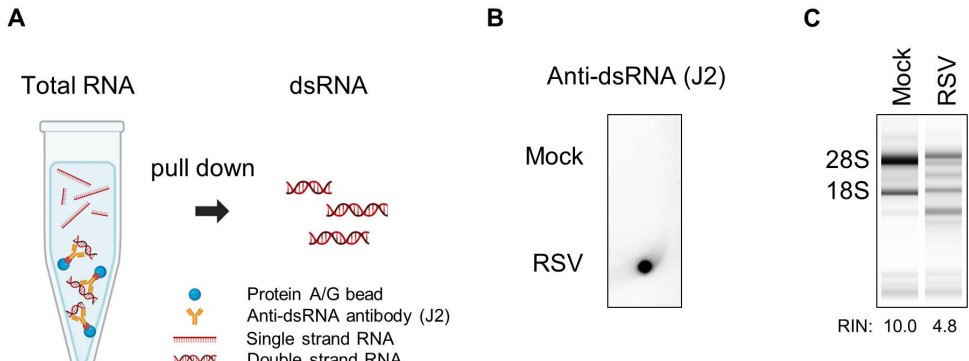

**Fig 6. Naked dsRNA isolated from RSV-infected cells induces OAS-RNase L activation. (A)** Total RNA extracted from the RSV- or mock-infected cells was used in a pull-down assay with anti-dsRNA antibody pre-bounded to protein A/G beads. **(B)** dsRNA enrichment was confirmed via dot blotting using an anti-dsRNA antibody (J2). **(C)** After the purified dsRNA was transfected into the A549 cells, total RNA harvested at 24 h post-transfection was analyzed using the RNA TapeStation System. Schematic overview for this experiment was created with BioRender.com. Created in BioRender. Kwon, **Y.** (2026) https://BioRender.com/unis62m.

site of infection and the first line of innate immune defense [56]. By preventing OAS-RNase L pathway activation, RSV can establish robust replication in these epithelial cells, leading to the exuberant neutrophilic inflammation and cytokine production that correlates with disease severity in RSV bronchiolitis. Such delayed viral clearance likely exacerbates epithelial injury and enhances immune-mediated pathology.

Unexpectedly, RSV infection did not activate the OAS-RNase L pathway, even though infection upregulated the expression levels of OAS-RNase L pathway components. Furthermore, attempts to increase OASs' sensitivity by overexpressing OAS isoforms did not result in pathway activation. Additionally, RSV did not halt the ectopic activation of the OAS-RNase L pathway but this activation still failed to inhibit RSV replication. Numerous studies have identified diverse strategies by which viruses evade the OAS-RNase L pathway, including the direct inhibition of RNase L enzymatic activity (e.g., Theiler's virus) and the structurally sequestration of viral dsRNA within viral proteins to prevent detection (e.g., Human immunodeficiency, Influenza A, and vaccinia viruses) [42,57–59]. Flaviviruses employ a distinct mechanism in which membrane-associated replication organelles create spatial compartmentalization that physically shields viral genomic RNA from RNase L-mediated cleavage, even when activated RNase L degrades cellular mRNAs. This membrane-based sequestration within replication complexes enables sustained viral replication despite robust pathway activation, representing a mechanism distinct from direct enzymatic inhibition or protein-mediated dsRNA shielding [39,43]. In this study, we have revealed a novel immune evasion strategy employed by RSV, in which LLPS, independent of membranes, sequesters viral dsRNA from the OAS-RNase L pathway.

The dsRNA, the initial trigger of the OAS-RNase L pathway, does not allow for the probing of RSV-infected cells via IFA using anti-dsRNA antibodies; however, it was successfully detected in the total RNA (Fig 3A and 3B). We focused on viral IBs as the primary structures responsible for dsRNA shielding. In RSV-infected cells, IB formation is characteristically observed as a droplet shape, which is enriched with viral proteins such as N, P, M2-1, and L that are linked to replication and transcription [20–24]. These IBs serve as the main sites of new viral RNA synthesis [25]. Moreover, RSV facilitates immune evasion by trapping immune factors, such as MDA5, MAVS, and NF-κB subunit p65, in the IBs [29,30]. The dense structural properties of IBs hinder antibodies from accessing internal epitopes, leading to the restriction of fluorescent signals to the IB periphery, complicating the exploration of their internal organization [25]. Thus, we hypothesized that dsRNA resides in IBs that act as a barrier to its detection by OASs and anti-dsRNA antibodies. Proteinase K treatment enabled detection of dsRNA co-localizing with IBs; notably, this dsRNA localized to IBs

but was excluded from IBAGs, a distribution pattern consistent with that previously reported for RSV genomic RNA [25]. M2-1 mediates trafficking of newly synthesized viral RNA to IBAGs; however, its low affinity for double-stranded RNA likely accounts for the exclusion of dsRNA from these substructures. Consequently, RSV-derived dsRNA, predominantly cbDVGs generated during viral genome replication, accumulates within IBs but outside of IBAGs. As an alternative possibility of dsRNA conceal, we considered that M2-1 might structurally shield short RNA duplexes (i.e., dsRNA-like structures) within the M2-1–SH7 RNA complex [60]. Structural analyses, however, indicate that M2-1 engages exclusively with unpaired RNA regions, leaving the duplex portion exposed, and Gao et al. further proposed that the observed base-pairing likely represents a crystallographic artifact rather than a physiologically relevant conformation.

We disrupted IBs by deleting viral genes necessary for their formation to investigate the ability of IBs to conceal dsRNA and their effect on activating the OAS-RNase L pathway. However, this approach proved unfeasible because the absence of N and P proteins or impaired IBs formation critically abrogated viral replication. Recent studies have identified IBs generated during RSV infection as biomolecular condensates formed via LLPS, similar to those observed in other negative-stranded RNA viruses [26,27]. Suppression of RSV replication after LLPS hardening by A3E or cyclopamine (CPM) treatment underscores LLPS's crucial role in RSV replication [28]. Surprisingly, treatment with 1,6-HD or hypotonic shock disrupted the condensation of IBs and induced dsRNA leakage into the cytosol. Notably, IBs hardening with CPM prevented hypotonic shock–induced dsRNA leakage but did not inhibit 1,6-HD–induced leakage (S10A and S10B Fig). These results show that RSV may use LLPS to sequester its dsRNA and evade sensing from OASs. Moreover, this masking strategy effectively prevents ectopically activated RNase L from degrading viral RNA. Contrary to our expectations, OAS-RNase L pathway activation was not observed in response to 1,6-HD-induced dsRNA leakage during RSV infection, possibly because of the pleiotropic effects of 1,6-HD. Düster et al. reported that 1,6-HD is unsuitable for investigating the functional relationship between LLPS and cellular pathways [61,62]. These results suggest that 1,6-HD impairs the activities of enzymes, including those of kinases and phosphatases, which are crucial for cellular signaling and function. The mechanism of action of 1,6-HD suggests that it exerts non-negligible effects on multiple pathways, including the OAS-RNase L pathway, as demonstrated in our study, although their hypothesis mainly focused on the enzymatic activity of kinases and phosphatases. We observed a similar pattern with hypotonic shock treatment, dsRNA was detected by IFA but did not activate the OAS-RNase L pathway. Notably, the intensity of dsRNA release induced by hypotonic shock was considerably lower than that achieved with 1,6-HD. Previous studies demonstrated that hypotonic shock-disrupted RSV IBs rapidly recover [28]. Collectively, these findings lead us to hypothesize that hypotonic shock induces only incomplete dsRNA exposure from IBs, thereby restricting the window for OAS-mediated sensing.

In summary, the OAS-RNase L pathway was not activated during RSV infection, and ectopically activated RNase L failed to suppress viral replication. The initial trigger of this pathway produced during RSV replication, dsRNA, was strictly sequestered within the LLPS-mediated IBs but liberated upon LLPS disruption. Notably, transfection with naked dsRNA isolated from RSV-infected cells, which eliminated the protective effect of LLPS, activated the OAS-RNase L pathway. Thus, RSV may evade the OAS-RNase L pathway by sequestering viral RNA via LLPS-driven IBs formation (Fig 7). These findings suggest that modulating the physical properties of LLPS structures, such as IBs, could represent a novel antiviral therapeutic strategy against RSV. In addition to hardening IBs, approaches aimed at specifically dissolving IBs to expose sequestered viral RNA to the immune system could be explored for their therapeutic potential [28]. This study is the first to demonstrate that LLPS acts as a shielding barrier that protects RNA from RNase-induced degradation. It is unlikely that this evasion strategy is restricted to the OAS-RNase L pathway and further research is required to determine whether LLPS similarly enables escape from other RNA-sensing immune pathway. Together, these findings indicate that RSV exploits LLPS-driven IBs to sequester viral dsRNA from OAS-RNase L pathway, thereby broadly dampening host antiviral responses.

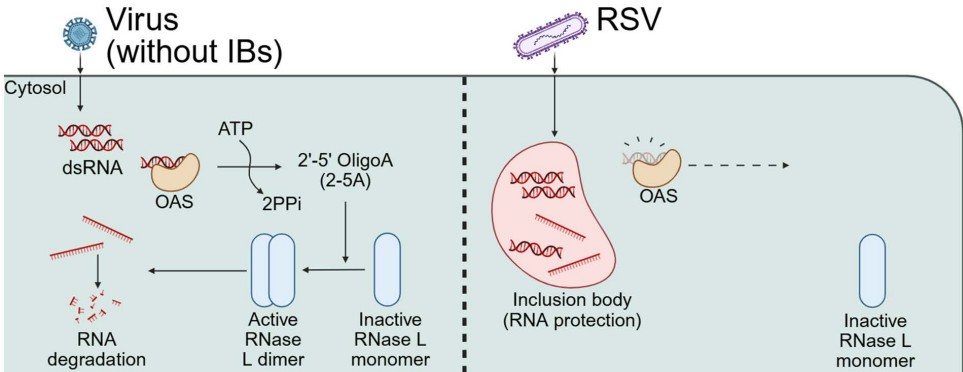

**Fig 7. Schematic model of RSV evasion from the dsRNA-mediated OAS-RNase L pathway.** (left) Upon viral dsRNA detection produced by diverse viruses, OASs bind to them and produce 2-5A, which activate RNase **L**. Cellular and viral single-strand RNAs are degraded, leading to restriction of virus replication and host cell apoptosis. (right) In RSV-infected cells, inclusion bodies (IBs) sequester replication-derived dsRNA, preventing OAS detection and pathway activation. Additionally, IBs shield the viral genome from apically activated RNase **L**. This schematic figure was created with BioRender. com. Created in BioRender. Kwon, **Y.** (2026) https://BioRender.com/etgj415.

## Materials and methods

### Cells and viruses

HEp-2 cells (10023, KCLB) were cultured in minimum essential medium supplemented with 100 U/mL penicillin–streptomycin (Gibco) and 10% (v/v) fetal bovine serum (FBS). A549 cells (ATCC CCL-185) were obtained from the American Type Culture Collection (ATCC; Manassas, VA, USA) and maintained in Dulbecco's modified Eagle's medium supplemented with 100 U/mL penicillin–streptomycin and 10% FBS. A549-RNase L KO stably expressing GFP-RNase L and mRuby2-OAS3 cells were kindly provided by Dr. Burke at UF Scripps [43,44]. NHBE (PCS-300–010, ATCC) cells were maintained in airway epithelial cell basal medium (PCS-300–030) supplemented with Bronchial Epithelial Cell Growth Kit (PCS-300–040). RSV strains A2 (VR-1540) and 18537 (VR-1580) were acquired from ATCC and amplified within the HEp-2 cells. Both viral strains were propagated three times before use in the experiments. Virus titers were determined via a plaque assay on the HEp-2 cells at 37°C using a methylcellulose overlay for 7 d after infection.

### rRNA cleavage assay

A549, HEp-2 and NHBE cells were infected with the indicated virus at an MOI ranging from 0.1–5 or transfected with poly (I:C) (Sigma). Cells were harvested using Trizol reagent (Ambion) at the indicated time points following infection or transfection. rRNA cleavage was analyzed using a 4150 TapeStation System (Agilent Technologies, California, USA) with an RNA ScreenTape (5067–5576, Agilent Technologies).

### Immunoblotting

Cells were lysed in radioimmunoprecipitation assay (RIPA) buffer (RC2002-050-00, Biosesang) for immunoblotting. Proteins were separated via electrophoresis on 10% denaturing polyacrylamide gels and transferred onto polyvinylidene difluoride membranes (Merck Millipore). The membranes were blocked with 5% skim milk (BD Biosciences, Franklin Lakes, NJ, USA) in Tris-buffered saline containing 0.1% Tween 20 for 1 h at 24°C. Subsequently, the membranes were incubated overnight with the following primary antibodies: anti-OAS1 (14498S, Cell Signaling Technology, 1:200), anti-OAS2 (54155S, Cell Signaling Technology, 1:500), anti-OAS3 (21915–1-AP-20, Proteintech, 1:250), anti-RNase L (27281S, Cell Signaling Technology, 1:1,000), anti-β-actin (sc-47778, Santa Cruz Biotechnology, 1:5,000), anti-RSV N (GTX636648, GeneTex, 1:1,000), and

anti-ZIKV NS3 (GTX133309, GeneTex, 1:1,000). Proteins were detected using horseradish peroxidase (HRP)-conjugated secondary antibodies (Bio-Rad) and an enhanced chemiluminescence (ECL) reagent (Thermo Fisher Scientific).

RNA dot blot assays to detect dsRNA in cell lysates involved isolating total RNA using Trizol reagent (Invitrogen) following the manufacturer's instructions, spotting onto nylon membranes (Sigma), and incubating overnight with 5% skim milk (BD Biosciences) in phosphate-buffered saline containing 0.05% Tween 20 (PBS-T) at 4°C. dsRNA was probed using the anti-dsRNA antibodies J2 (76651, Cell Signaling Technology, 1:1,000) and 9D5 (Ab00458–23, Absolute antibody, 1:1,000) for 2 h at 24°C. For dsRNA visualization, HRP-conjugated secondary antibodies (Bio-Rad) and ECL reagents (Thermo Fisher Scientific) were used.

### Quantitative reverse transcription polymerase chain reaction (RT-PCR)

RT-qPCR (QuantStudio 3, Applied Biosystems, Foster City, CA, USA) was performed using One-Step PrimeScript III RT-qPCR Mix (Takara Bio, Shiga, Japan). The RSV F gene was detected using a probe-based qPCR assay (Integrated DNA Technologies, Coralville, IA, USA). Additionally, the *IFN-β*, *OAS1*, *OAS2*, and *OAS3* genes were detected using individual customized probes (Integrated DNA Technologies). S1 Table lists the sequences of the qPCR probes and primers used in this study.

### Immunofluorescence staining

The cells were seeded onto 8-chamber slides and cultured overnight. Cells were infected with RSV A2 at an MOI of 2, fixed with 4% paraformaldehyde for 20 min at the indicated time points post-infection, permeabilized using 0.1% Triton X-100 in PBS-T for 20 min, and blocked with 5% bovine serum albumin for 1 h at 24°C. The slides were incubated overnight with the following primary antibodies: anti-dsRNA antibodies J2 and 9D5, anti-RSV N, anti-RSV F (01-07-0121, Cambridge, 1:1,000), and anti-RSV P (ab94965, Abcam, 1:1,000). Subsequently, the slides were rinsed three times with PBS-T and incubated with the appropriate Alexa Fluor-conjugated secondary antibodies (Invitrogen) for 1 h at 24°C. For nuclear staining, 4',6-diamidino-2-phenylindole (DAPI) was used.

To disrupt LLPS, RSV- or mock-infected cells were treated at 24 h post-infection with 5% 1,6-HD prepared in MEM or subjected to hypotonic shock using 10% hypotonic medium in distilled water for the indicated durations. After treatment, the cells were rinsed three times and fixed. Immunofluorescence staining was performed as previously described. For LLPS hardening, 5 μM CPM (HY-17024–1mg, MCE) was treated for 1 h before LLPS disruption.

### Naked dsRNA purification from RSV-infected cells and transfection

Naked dsRNA from RSV-infected cells was isolated using a previously reported protocol, with some modifications [49]. Briefly, four 150-mm dishes of HEp-2 cells were either mock infected or infected with RSV A2 at an MOI of 2. After 48 h, the total RNA was extracted using Trizol reagent. For dsRNA enrichment, anti-dsRNA monoclonal antibody J2 pre-conjugated to protein A/G magnetic beads (20424, Thermo) was incubated with total RNA in a pre-chilled reaction buffer (20 mM Tris-HCl [pH 7.0], 150 mM NaCl, 0.2 mM EDTA, 0.2% Tween-20) at 4°C for 2 h with rotational mixing. Using a microspin column (89879, Thermo), beads were recovered and washed three times with wash buffer (20 mM Tris-HCl [pH 7.0], 150 mM NaCl, 0.1 mM EDTA, 0.1% Tween-20). After resuspending the beads in 150 μL of wash buffer, bound dsRNA was eluted, followed by the addition of 450 μL Trizol LS reagent (10296010, Invitrogen). Finally, the dsRNA was isolated according to the manufacturer's instructions. The dsRNA isolated from RSV- or mock-infected cells was analyzed by RNA dot blotting as previously described and subsequently transfected into A549 cells in equal volumes.

### Statistical analysis

All experiments were conducted at least three times. All data were analyzed using GraphPad Prism 8.0 (GraphPad Software, San Diego, CA, USA). Statistical significance was set at $P < 0.05$. The figure legends include the description of the specific analytical methods.

## Supporting information

**S1 Fig. RSV infection fails to activate the OAS–RNase L pathway, whereas ZIKV infection triggers.** (A, B) HEp-2 cells were infected with RSV A2 at an MOI of 2 and lysed at the indicated time points. rRNA cleavage and RNA integrity number (RIN) were analyzed using the RNA TapeStation System. Viral replication was examined by immunoblotting with anti-RSV N and anti-β-actin antibodies. (C, D) The A549 cells were infected with ZIKV PRVABC59 at an MOI of 2. RNA and protein were harvested and examined using the RNA TapeStation System and immunoblotting with anti-ZIKV NS3 and anti-β-actin antibodies.
(DOCX)

**S2 Fig. RNase L was not activated in the RSV-infected A549 cells overexpressing OAS isoforms.** The A549 cells were transfected with the OASs isoforms 24 h before RSV infection at an MOI of 2. At 2 d post-infection, the RNA was purified and analyzed using the RNA TapeStation System.
(DOCX)

**S3 Fig. The dsRNA detection in total RNA extracted from the RSV-infected cells.** (A) overview about detecting dsRNA through IFA and dot-blotting. (B) Total RNA from the mock-, RSV-, and ZIKV-infected cells was transferred onto nylon membranes. The dsRNA was detected using an anti-dsRNA antibody (9D5). Schematic overviews for these experiments were created with BioRender.com. Created in BioRender. Kwon, Y. (2026) https://BioRender.com/u5t1cmx.
(DOCX)

**S4 Fig. Progressive growth of RSV inclusion bodies and recruitment of N, P, and M2-1 proteins.** (A) A549 cells were infected with RSV A2 at an MOI of 2 and stained with anti-RSV N antibody (green) at the indicated time points. (B) A549 cells were infected with RSV A2 at an MOI of 2 and probed with anti-RSV N antibody (green), anti-RSV P (red) and anti-RSV M2-1 (red) indicated time points. Scale bar, 20 μm.
(DOCX)

**S5 Fig. OAS3 and RNase L were excluded from inclusion bodies (IBs).** (A) A549-RNase L KO stably expressing GFP-RNase L and mRuby2-OAS3 cells were infected with RSV A2 at an MOI of 2, stained with anti-N antibody (green) at 24 h post-infection, and imaged using an LSM 980 confocal microscope. (B) The white line indicated the rack of a line intensity profile. Scale bar, 10 μm.
(DOCX)

**S6 Fig. Disruption of inclusion bodies (IBs) with an LLPS inhibitor induces dsRNA leakage into the RSV-infected cells.** The A549 cells were infected with RSV A2 at an MOI of 2. Before the cells were fixed 24 h after infection, they were treated with 5% 1,6-HD at the indicated times. Cells were stained with anti-RSV P antibody (green) and anti-dsRNA (9D5, red). Scale bar, 20 μm.
(DOCX)

**S7 Fig. 1,6-hexanediol (1,6-HD) or hypotonic shock do not induce host dsRNA.** (A–B) A549 and NHBE cells were exposed to 1,6-HD or hypotonic shock, stained with anti-dsRNA antibody (red), and imaged using an LSM 980 confocal microscope. Scale bar, 10 μm.
(DOCX)

**S8 Fig. LLPS disruption does not induce rRNA cleavage during RSV infection.** (A–B) A549 and NHBE cells were infected with RSV A2 at an MOI of 2. At 24 h post-infection, cells were exposed to 1,6-HD or hypotonic shock for 5 min, followed by a 24 h recovery period. Total RNA was then harvested and analyzed using the RNA TapeStation System.
(DOCX)

**S9 Fig. rRNA cleavage by transfection of the total RNA extracted from the RSV-infected cells and Naked dsRNA purified from the RSV-infected cells.** (A) The total RNA was purified from mock- and RSV-infected cells and transfected into A549 cell with 10 μg. At 24 h after-transfection, the RNA was assessed using the RNA TapeStation System. (B) After dsRNA purification, dsRNA enrichment was confirmed by dot blotting using an anti-dsRNA antibody (9D5).
(DOCX)

**S10 Fig. Cyclopamine (CPM) blocks hypotonic shock–induced dsRNA release but not 1,6-hexanediol (1,6-HD).** (A–B) A549 and NHBE cells were infected with RSV A2 at an MOI of 2. At 24 h post-infection, cells were treated with 5 μM CPM for 1 h and then exposed to 1,6-HD or hypotonic shock for 5 min, followed by staining with anti-RSV N antibody (green) and anti-dsRNA (J2, red). Scale bar, 10 μm.
(DOCX)

**S1 Table. qRT-PCR primers and probes.**
(DOCX)

## Acknowledgments

We would like to express our special thanks to Dr. Byung-Yoon Ahn (Korea University) for his valuable support, and to Dr.Burke (UF Scripps, Jupiter FL, USA) for generously providing the A549-RNase L KO stably expressing GFP-RNase L and mRuby2-OAS3 cells used in this study.

## Author contributions

**Conceptualization:** Woo Yeon Hwang, Soohwan Oh, Young-Chan Kwon.

**Data curation:** Woo Yeon Hwang, Young-Chan Kwon.

**Funding acquisition:** Soohwan Oh, Young-Chan Kwon.

**Investigation:** Woo Yeon Hwang.

**Methodology:** Woo Yeon Hwang, Soohwan Oh, Young-Chan Kwon.

**Supervision:** Soohwan Oh, Young-Chan Kwon.

**Visualization:** Woo Yeon Hwang.

**Writing – original draft:** Woo Yeon Hwang, Soohwan Oh, Young-Chan Kwon.

**Writing – review & editing:** Michael G. Rosenfeld, Soohwan Oh, Young-Chan Kwon.

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
