## [Decision Letter · Decision Letter 0]

24 Sep 2025

Liquid-liquid phase separation mediated immune evasion of respiratory syncytial virus against oligoadenylate synthetase-RNase L pathway

PLOS Pathogens

Dear Dr. Kwon,

Thank you for submitting your manuscript to PLOS Pathogens. After careful consideration, we feel that it has merit but does not fully meet PLOS Pathogens's publication criteria as it currently stands. Therefore, we invite you to submit a revised version of the manuscript that addresses the points raised during the review process.

Please submit your revised manuscript within 60 days Nov 23 2025 11:59PM. If you will need more time than this to complete your revisions, please reply to this message or contact the journal office at plospathogens@plos.org. Please include the following items when submitting your revised manuscript:

We look forward to receiving your revised manuscript.

Kind regards,

Barry Rockx

Academic Editor

PLOS Pathogens

Thomas Hoenen

Section Editor

PLOS Pathogens

Editor-in-Chief

PLOS Pathogens

orcid.org/0000-0003-2946-9497

Editor-in-Chief

PLOS Pathogens

orcid.org/0000-0002-7699-2064

**Journal Requirements:**

At this stage, the following Authors/Authors require contributions: Woo Yeon Hwang, Michael G. Rosenfeld, Soohwan Oh, and Young-Chan Kwon. Please ensure that the full contributions of each author are acknowledged in the "Add/Edit/Remove Authors" section of our submission form.

3) We notice that your supplementary Figures, and Tables are included in the manuscript file. Please remove them and upload them with the file type 'Supporting Information'. Please ensure that each Supporting Information file has a legend listed in the manuscript after the references list.

Potential Copyright Issues:

- Figures 3A and 6A. Please confirm whether you drew the images / clip-art within the figure panels by hand. If you did not draw the images, please provide (a) a link to the source of the images or icons and their license / terms of use; or (b) written permission from the copyright holder to publish the images or icons under our CC BY 4.0 license. Alternatively, you may replace the images with open source alternatives. See these open source resources you may use to replace images / clip-art:

5) When completing the data availability statement of the submission form, you indicated that you will make your data available on acceptance. We strongly recommend all authors decide on a data sharing plan before acceptance, as the process can be lengthy and hold up publication timelines. Please note that, though access restrictions are acceptable now, your entire data will need to be made freely accessible if your manuscript is accepted for publication. This policy applies to all data except where public deposition would breach compliance with the protocol approved by your research ethics board. If you are unable to adhere to our open data policy, please kindly revise your statement to explain your reasoning and we will seek the editor's input on an exemption. Please be assured that, once you have provided your new statement, the assessment of your exemption will not hold up the peer review process.

6) Please ensure that the funders and grant numbers match between the Financial Disclosure field and the Funding Information tab in your submission form. Note that the funders must be provided in the same order in both places as well.

**Reviewers' Comments:**

Reviewer's Responses to Questions

**Part I - Summary**

Reviewer #1: The manuscript from Hwang et el. describes how RSV facilitates LLPS driven IBs in structurally sequestering viral RNA facilitating evasion of RNase dependent genomic RNA degradation by OAS-RNase L antiviral pathway. The study is well planned, backed up by sufficient supporting data, and clear illustration of results. Overall, the manuscript is well written, and the results are logically interpreted. The authors may please consider the following comments for minor clarification.

Specific Comments:

1. The authors need to briefly discuss physiological relevance with respect to the nature of immune evasion with respect to the outcome from respiratory immune response.

2. P. 5, line 4: What is meant by wild-type A549 cells? Line 13: The S2Fig needs little more clarification to match the statement from the results.

3. P. 7, 2nd paragraph: Rewarding the subheading of this paragraph will make the concluding result in line.

4. P. 9, Line 11: Why did anti-dsRNA (J2 Red) not detect RSV dsRNA?

5. Fig. 3, panel A and Fig. 6, panel A may appear little excess to some readers, although the schematics at a glance help to clarify the experimental approaches. The authors may add appropriate wordings in text for justification of the separate panels. Some of the supplementary figures could have been brought in context to the main result areas.

6. Fig. 6, panel B and S6B: Why do the blots show hollow vesicular staining by anti-dsRNA?

7. The impact of studied immune evasion pathway in context to the other immune regulatory mechanisms may be briefly discussed.

Reviewer #2: In this manuscript, Hwang and colleagues investigate how Respiratory syncytial virus (RSV) prevents the activation of the OAS-RNase L pathway. They show that RSV-infected cells have increased dsRNA by dot blot analyses but do not display RNase L activation based on the lack of RNase L-dependent of ribosomal RNA cleavage. The authors show that dsRNA is sequestered into inclusion bodies (IBs), which are susceptible to 1,6-hexanediol (1,6-HD) indicating that the IBs are liquid-liquid phase separation. Re-introduction of RNA isolated from RSV-infected cells can activate RNase L. Combined, the authors argue that RSV dsRNA replication intermediates are sequestered within IBs, and that this limits the ability of RNase L to activate.

Overall, the data support their conclusions, and the manuscript is well-written. While it is already known that many negative strand RNA viruses do not activate RNase L, and it has long been assumed that inclusion bodies prevent sensing of viral RNA substrates by pattern recognition receptors such as OAS proteins, this study formally attempts to address this, which is impactful to the field of intracellular innate immunology and virology. However, a major limitation of this study is that the authors do not provide a mechanism by which the RSV IBs prevent RNase L activation. To increase the impact of this study, the authors should consider exploring the possibility that the IBs exclude OAS and RNase L proteins, thus limiting the ability of OAS protein and RNase L to condense on viral dsRNA and activated (Cusic and Burke, Sci Signal. 2024 PMID: 38771918; Briggs et al., Genes Dev. 2025 PMID: 40588419). The authors should consider showing that OAS3 and RNase L are excluded from RSV IBs by fluorescent microscopy, and that the interaction between dsRNA and OAS proteins by proximity labeling assay (PLA) does not occur in IBs.

This review was written by James Burke (University of Florida Scripps Institute), and I am happy to provide any reagents that would be helpful (A549 cells that stably express GFP-RNase L, RFP-OAS3, etc).

Minor comments:

1. Does 1,6-hexanediol (1,6-HD) treatment lead to RNase L activation? This would be a powerful experiment that strongly supports that the IBs prevent RNase L activation.

2. Fig. S7. RNA from mock-infected cells seemed to activate RNase L to some level. The RNA from RSV-infected cells did not lead to robust RNase L activation. It is unclear if dsRNA is host or viral, as RSV infection could lead to host dsRNA accumulation that could lead to RNase L activation. Does pre-expression of N block RNase L from RSV-infected cell derived RNA?

3. Briggs et al. 2025 (PMID: 40588419) showed that flavivirus RNA evades the effects of RNase L-mediated RNA decay, whereby RNase L can activate and degrade all cellular mRNA but viral RNA is not degraded. The authors should consider including this in their discussion on the role of viral replication complexes in evading the RNase L pathway.

Reviewer #3: This manuscript addresses how respiratory syncytial virus (RSV) inclusion bodies (IBs) formed by liquid–liquid phase separation (LLPS) enable immune evasion from the OAS–RNase L pathway. The authors show that RSV infection does not activate RNase L despite dsRNA production (Results, p.5–7, Fig 1), that ectopically activated RNase L fails to inhibit replication (Results, p.7–8, Fig 2), and that dsRNA is sequestered within IBs but becomes detectable after IB disruption with 1,6-hexanediol (Results, p.13–14, Fig 5). Purified dsRNA from infected cells activates the OAS–RNase L pathway (Results, p.15, Fig 6).

Strengths

• Logical experimental flow from lack of pathway activation → dsRNA sequestration → functional shielding.

• Convergent evidence from imaging and biochemical assays.

• Important conceptual contribution: IBs serve as both replication factories and immune-evasion compartments.

Weaknesses

• Reliance on 1,6-hexanediol as the sole LLPS perturbant.

• No validation in IFN-competent primary cells.

• Lack of discussion of IBAGs, central to RSV RNA trafficking.

• No distinction between genomic RNA, replication intermediates, and viral mRNAs.

• No acknowledgment of structural shielding mechanisms already shown by Gao et al., Structure, 2020/2021.

• Some overstatements of novelty (e.g., “IBs are viral factories,” which is long established).

Overall, the manuscript is carefully executed and of potential interest, but additional mechanistic validation and integration with known RNA biology are required before publication.

**Part II – Major Issues: Key Experiments Required for Acceptance**

Please use this section to detail the key new experiments or modifications of existing experiments that should be absolutely required to validate study conclusions.required to validate study conclusions.required to validate study conclusions.required to validate study conclusions.

Reviewer #1: The manuscript needs some minor clarifications.

I have clarified the issues raised in my mind in the Summary/Sp. Comments section. The authors may choose to move some supplemental figures to the results section and I would suggest the authors to decide.

Reviewer #2: The authors should provide a mechanism by which dsRNA sequestration in IBs prevent RNase L activation. For example, are OAS3 and RNase L excluded from RSV-IBs?

Does 1,6-hexanediol (1,6-HD)-mediated dissolution of RSV-IBs lead to RNase L activation?

Reviewer #3: The key evidence for dsRNA shielding is derived from 1,6-hexanediol (1,6-HD) treatment (Results, p.13–14, Fig 5). However, 1,6-HD has broad off-target effects and may directly impair OAS–RNase L activity, confounding interpretation (Discussion, p.17). Prior work (Risso-Ballester et al., Nature, 2021) demonstrated that CPM-hardened IBs dissolve with 1,6-HD but resist hypotonic shock. I strongly recommend performing a 1,6-HD vs hypotonic shock ± CPM matrix, with readouts for IB material state (FRAP, morphology, p.11–12, Fig 4), dsRNA localization (IFA, p.11–13, Fig 4C), OAS–RNase L activation (rRNA cleavage, p.5–7, Fig 1; 2-5A; RNase L dimerization), and replication (Fig 1C, Fig 2B). Poly(I:C) transfection controls (Results, p.7–8, Fig 2A) should confirm pathway competence. This would firmly link IB material state to dsRNA shielding.

2. Validation in an IFN-competent system.

All data are from A549 and HEp-2 cells (Methods, p.20–21), which have attenuated innate immune signaling. At least the key ± CPM, ± hypotonic shock dataset should be reproduced in primary human airway epithelial cells (NHBE or ALI cultures). Even a limited dataset here would provide crucial physiological relevance.

3. Incomplete treatment of RNA biology and structural mechanisms.

The manuscript convincingly shows dsRNA shielding but omits critical aspects of RSV RNA biology:

• IBAGs omitted: Inclusion body–associated granules (IBAGs) are the recognized substructures where viral mRNAs accumulate and exit IBs (Rincheval et al., Nat Commun 2017). Their absence from the Introduction, Results (p.11–12, Fig 4), and Discussion (p.16–18) is striking.

• RNA species not distinguished: The dsRNA detected (Results, p.9–11, Fig 3C) could represent genomic RNA, replication intermediates, or mRNA duplexes. The authors should at least discuss this limitation, and ideally provide RNA FISH or qPCR data for viral mRNAs.

• Structural shielding not discussed: Gao et al., Structure, 2020/2021 demonstrated that RSV gene-end ds mRNA is directly occluded by M2-1/P–RNA complexes. This structural shielding mechanism should be acknowledged and integrated into the Discussion (p.16–18).

Without addressing these points, the mechanistic framework risks being oversimplified as LLPS-only shielding.

**Part III – Minor Issues: Editorial and Data Presentation Modifications**

Reviewer #1: The authors describes how cleverly RSV manipulates innate immune system of the host to attenuate a prompt and robust antiviral immunity. The manuscript describes a nice piece of intelligent work.

Reviewer #2: None

Reviewer #3: (No Response)

PLOS authors have the option to publish the peer review history of their article (what does this mean?). If published, this will include your full peer review and any attached files.). If published, this will include your full peer review and any attached files.). If published, this will include your full peer review and any attached files.). If published, this will include your full peer review and any attached files.

...

Reviewer #1: No

Reviewer #2: **Yes:** James M. BurkeJames M. BurkeJames M. BurkeJames M. Burke

Reviewer #3: No

**Figure resubmission:**

**Reproducibility:**



---

## [Decision Letter · Decision Letter 1]

17 Mar 2026

Dear Dr. Kwon,

We are pleased to inform you that your manuscript 'Liquid-liquid phase separation mediated immune evasion of respiratory syncytial virus against oligoadenylate synthetase-RNase L pathway' has been provisionally accepted for publication in PLOS Pathogens.

Best regards,

Barry Rockx

Academic Editor

PLOS Pathogens

Thomas Hoenen

Section Editor

PLOS Pathogens

Sumita Bhaduri-McIntosh

Editor-in-Chief

PLOS Pathogens

orcid.org/0000-0003-2946-9497

Michael Malim

Editor-in-Chief

PLOS Pathogens

orcid.org/0000-0002-7699-2064

Reviewer Comments (if any, and for reference):

Reviewer's Responses to Questions

**Part I - Summary**

Reviewer #1: The revised manuscript reads much improved. The authors addressed all my points in the revised version of the manuscript.

Reviewer #2: (No Response)

Reviewer #3: The authors have carefully addressed the major concerns raised in the previous review. In particular:

• The exclusion of OAS3 and RNase L from RSV inclusion bodies is now experimentally demonstrated.

• The comparison of 1,6-hexanediol and hypotonic shock conditions strengthens the interpretation that LLPS contributes to dsRNA sequestration.

• Key experiments have been reproduced in primary bronchial epithelial cells, improving physiological relevance.

• The manuscript now integrates IBAG biology and more clearly discusses the nature of the dsRNA species detected.

• Structural shielding mechanisms reported previously (e.g., Gao et al.) are acknowledged and discussed, reducing earlier conceptual oversimplification.

While some mechanistic questions remain open — particularly regarding the precise molecular interface governing dsRNA retention and the quantitative threshold required for OAS activation — these do not preclude publication.

Overall, the revised manuscript now presents a coherent and well-supported model in which LLPS-mediated inclusion bodies function as immune-evasion compartments that spatially restrict access of the OAS–RNase L machinery to viral dsRNA.

I believe the authors have sufficiently addressed the prior concerns and that the manuscript is suitable for publication in PLOS Pathogens.

**Part II – Major Issues: Key Experiments Required for Acceptance**

Please use this section to detail the key new experiments or modifications of existing experiments that should be absolutely required to validate study conclusions.required to validate study conclusions.required to validate study conclusions.required to validate study conclusions.

Reviewer #1: None

Reviewer #2: The authors adequately addressed reviewer comments. However, I suggest the authors make some minor changes before publication.

1. The data in Figure S5 are vital for their overall model that IBs exclude OAS3/RNase L from dsRNA. Without these data, their model is unjustified and unsupported by data. Thus, I strongly suggest moving these data to the main figure. In addition, provide plot profile analyses, whereby the intensity of dsRNA, OAS3, and RNase L is plotted along the distance spanning the IB to show that RNase L and OAS3 is excluded from the IB. Then show the average depletion from at least 10 cells.

2. Please acknowledge the origin of the A549 mRuby-OAS3 and GFP-RNase L cell lines in the acknowledgment section, as this is important for NIH progress reports.

Reviewer #3: no additional experiments required

**Part III – Minor Issues: Editorial and Data Presentation Modifications**

Reviewer #1: None

Reviewer #2: (No Response)

Reviewer #3: no further modifications needed

PLOS authors have the option to publish the peer review history of their article (what does this mean?). If published, this will include your full peer review and any attached files.). If published, this will include your full peer review and any attached files.). If published, this will include your full peer review and any attached files.). If published, this will include your full peer review and any attached files.

...

Reviewer #1: No

Reviewer #2: No

Reviewer #3: **Yes:** Dr. Ralf AltmeyerDr. Ralf AltmeyerDr. Ralf AltmeyerDr. Ralf Altmeyer

---

## [Editor Report · Acceptance letter]

Dear Dr. Kwon,

We are delighted to inform you that your manuscript, "Liquid-liquid phase separation mediated immune evasion of respiratory syncytial virus against oligoadenylate synthetase-RNase L pathway," has been formally accepted for publication in PLOS Pathogens.

Best regards,

Sumita Bhaduri-McIntosh

Editor-in-Chief

PLOS Pathogens

orcid.org/0000-0003-2946-9497

Michael Malim

Editor-in-Chief

PLOS Pathogens

orcid.org/0000-0002-7699-2064